# New Insights from Soil Microorganisms for Sustainable Double Rice-Cropping System with 37-Year Manure Fertilization

Jin Li [1,2,†], Kai-Luo Liu [3,†], Ji Chen [4,5,6,*], Jiang Xie [2], Yu Jiang [7], Guo-Qiang Deng [2], Da-Ming Li [3], Xian-Jiao Guan [2], Xi-Huang Liang [2,8], Xian-Mao Chen [2,8], Cai-Fei Qiu [2], Yin-Fei Qian [2], Wen-Jian Xia [2], Jia Liu [2], Chun-Rui Peng [2], Stephen M. Bell [9] and Jin Chen [2,8,*]

1   School of Tourism and Economic Management, Nanchang Normal University, Nanchang 330032, China
2   Soil and Fertilizer & Resources and Environment Institute, Jiangxi Academy of Agricultural Sciences/Key Laboratory of Crop Ecophysiology and Farming System for the Middle and Lower Reaches of the Yangtze River, Ministry of Agriculture and Rural Affairs/National Engineering and Technology Research Center for Red Soil Improvement/National Agricultural Experimental Station for Agricultural Environment of Yichun, Nanchang 330200, China
3   Jiangxi Institute of Red Soil/ National Engineering and Technology Research Center for Red Soil Improvement, Nanchang 330046, China
4   Department of Agroecology, Aarhus University, Blichers Allé 20, 8830 Tjele, Denmark
5   Aarhus University Centre for Circular Bioeconomy, Aarhus University, 8830 Tjele, Denmark
6   iCLIMATE Interdisciplinary Centre for Climate Change, Aarhus University, 4000 Roskilde, Denmark
7   Jiangsu Collaborative Innovation Center for Modern Crop Production/National Engineering and Technology Center for Information Agriculture/Key Laboratory of Crop Physiology and Ecology in Southern China, Nanjing Agricultural University, Nanjing 210095, China
8   Jinggangshan Institute of Red Soil/ Jinggangshan Branch of Jiangxi Academy of Agricultural Sciences, Ji'an 343016, China
9   Laboratoire des Sciences du Climat et de l'Environnement, LSCE-IPSL (CEA-CNRS-UVSQ), Université Paris-Saclay, 91191 Gif-sur-Yvette, France
*   Correspondence: ji.chen@agro.au.dk (J.C.); chenjin2004777@163.com (J.C.);
    Tel.: +86-791-8783-2356 (J.C.); +45-7137-4531 (J.C.)
†   These authors contributed equally to this work.

**Abstract:** Long-term intensive use of mineral fertilizers in double rice-cropping systems has led to soil acidification and soil degradation. Manure fertilization was suggested as an alternative strategy to mitigate soil degradation. However, the effects of long-term mineral and manure fertilization on rice grain yield, yield stability, soil organic carbon (SOC) content, soil total nitrogen (TN) content, and the underlying mechanisms are unclear. Based on a long-term experiment established in 1981 in southern China, we compared four treatments: no fertilizer application (Control); application of nitrogen–phosphorus–potassium (NPK); NPK plus green manure in early rice (M1); and M1 plus farmyard manure in late rice and rice straw return in winter (M2). Our results showed that 37 years of NPK, M1, and M2 significantly increased rice grain yield by 54%, 46%, and 72%, and yield stability by 22%, 17%, and 9%, respectively. M1 and M2 significantly increased SOC content by 39% and 23% compared to Control, respectively, whereas there was no difference between Control and NPK. Regarding soil TN content, it was significantly increased by 8%, 46%, and 20% by NPK, M1, and M2, respectively. In addition, M2 significantly increased bacterial OTU richness by 68%, Chao1 index by 79%, and altered the bacterial community composition. Changes in soil nutrient availability and bacterial Simpson index were positively correlated with the changes in grain yield, while shifts in bacterial community were closely related to yield stability. This study provides pioneer comprehensive assessments of the simultaneous responses of grain yield, yield stability, SOC and TN content, nutrient availability, and bacterial community composition to long-term mineral and manure fertilization in a double rice-cropping system. Altogether, this study spanning nearly four decades provides new perspectives for developing sustainable yet intensive rice cultivation to meet growing global demands.

**Keywords:** organic amendment; double rice-cropping systems; bacterial community; reddish paddy soil; soil nutrient; sustainable agriculture

## 1. Introduction

Demand for rice (*Oryza sativa* L.) grain yield is anticipated to increase drastically over the coming decades with increasing global population [1–3]. To meet the rapidly growing demands on rice grain yield, land managers have been exploring various intensive rice cropping systems [4]. One example of this is the double rice-cropping system in South Asia whereby farmers grow two kinds of rice within one year and potentially double the yield produced from the same land area [5]. While long-term intensive double rice-cropping systems using mineral fertilization may cause soil acidification, soil degradation, and reductions in soil fertility [6,7], organic amendments such as manure and straw incorporation have been recognized as an effective management practice to maintain soil fertility and rice yield [8]. However, the long-term effects of mineral and manure fertilization on rice grain yield and their underlying mechanisms in double rice-cropping systems have not yet been comprehensively investigated based on the same experimental platform. This limits our abilities to recommend mineral, manure, or combination of the two as fertilizers to meet rice grain demands through double rice-cropping practices.

Apart from the grain yield, yield stability is an important proxy for the interannual variability of rice grain yield [9,10]. A higher yield stability indicates a greater capability to adapt to unfavorable environmental conditions, such as climate change or hydrological fluctuations [11]. Therefore, yield stability is closely linked to food security over the long-term [12,13]. The concept of yield stability was initially developed for plant breeding research [14], and has been insufficiently investigated in the context of double rice-cropping systems. Given the challenges of maintaining food security under a changing climate, sustainable double rice-cropping systems need to be designed not merely for larger grain yield, but also for higher yield stability [11]. However, grain yield and yield stability are rarely investigated simultaneously in double rice-cropping systems that incorporate mineral and manure fertilization practices.

Practices that promote sustainable rice grain yield and climate change mitigation goals imply the protection of soil organic carbon (SOC) and total nitrogen (TN) contents [15,16]. This is especially critical considering the high vulnerability of SOC and TN in rice cropping systems under rapid climate change [17]. For example, even minor changes in SOC and TN content can substantially increase greenhouse gas emissions and accelerate climate change [18,19]. The net effects of mineral and manure fertilization on SOC and TN content is still under debate. In the context of sustainable rice production, increasing of grain yield and yield stability should not be at the expense of reducing SOC and TN content [20,21], but the relationships between them remain unresolved. Therefore, before any broad recommendations of mineral or manure fertilization for sustainable double rice-cropping systems can be made, it is first necessary to better understand their effects on SOC and TN content and their relationships with grain yield and yield stability.

The effects of mineral and manure fertilization on grain yield, yield stability, SOC, and TN content are regulated by a range of different biotic and abiotic factors, and the relative contribution of each are unclear [20–22]. It has been reported that mineral and manure fertilization have contrasting effects on soil pH and soil EC [23], whereas their linkages with grain yield, yield stability, and SOC and TN content are still being investigated. Nutrient loadings may either increase or decrease soil available nutrient contents depending on the balance between assimilation, adsorption, and leaching [24,25], but again the net effects of mineral and manure fertilization on soil nutrient availability and their implications for sustainable double rice-cropping systems requires more dedicated research. Emerging studies provide some of the most compelling evidence that fertilization-induced shifts in soil microbial community composition and microbial biomass are closely related to the

changes in grain yield and SOC and TN content reported in other cropping systems [26,27]. To the best of our knowledge, there has been no study yet of double rice-cropping systems that has comprehensively and simultaneously investigated the effects of mineral and manure fertilization on grain yield, yield stability, SOC, and TN content, as well as the underlying mechanisms.

Long-term continuous field experimental platforms provide unique and valuable opportunities to better explore critical research questions that have previously only been addressed in the short-term due to funding or administrative constraints [28–31]. Moreover, the results from short-term studies are not always in line with long-term observations [22,32]. Therefore, we established a mineral and manure fertilization experimental platform in a double rice-cropping system in Southern China in 1981 to fully explore the effects of long-term mineral and manure fertilization on rice grain yield, yield stability, SOC, and TN content. Three research questions motivated this study. First, what are the effects of long-term mineral and manure fertilization on rice grain yield, yield stability, and SOC and TN content? Second, what are the effects of long-term mineral and manure fertilization on soil pH, soil EC, available nutrient content, and bacterial community composition? Third, what are the underlying mechanisms associated with changes in rice grain yield, yield stability, and SOC and TN content?

## 2. Experimental Procedures

### 2.1. Site Description

The long-term experimental platform was initiated in 1981 in Jinxian County of Jiangxi Province, China (116°17′55″ E, 28°35′38″ N, 21 m above sea level, Figure S1), and is managed by the Jiangxi Institute of Red Soil. The climate is subtropical monsoon humid. The average annual temperature, total sunshine hours, and length of frostless season are 18.1 °C, 1950 h, and 262 d, respectively. The monthly mean temperature reaches its lowest (4.6 °C) in January and highest (29.8 °C) in July (Figure S2). The annual precipitation is 1537 mm with about 38% falling during the early rice cultivation (from March to early July) and 14% during the late rice cultivation (from late July to early November). The soil developed from Quaternary red clay and was classified as a typic Stagnic Anthrosol based on the International Union of Soil Sciences classification system, which is a typical soil type in Southern China. Before the establishment of the experiment platform, soil pH was 6.9, soil organic matter was 27.96 g kg$^{-1}$, TN was 0.95 g kg$^{-1}$, total P was 1.02 g kg$^{-1}$, available N was 143.7 mg kg$^{-1}$, available P was 10.3 mg kg$^{-1}$, and available K was 125.1 mg kg$^{-1}$. More detailed information about the study site and the experimental design can be found in Bi et al. (2009) and Sun et al. (2013).

### 2.2. Experiment Design

A double rice-cropping system was selected, which includes early rice (April to July), late rice (July to October) and fallow or milk vetch (October to next April). Before the start of the experiment, the study site has been cultivated for rice for at least 100 years [16,31]. The four experimental treatments for this study were (Figure S1): (1) Control, no fertilizer; (2) NPK, mineral NPK fertilizer; (3) M1, NPK plus milk vetch manure in early rice; (4) M2, M1 plus farmyard manure in late rice and rice straw incorporation during milk vetch. The detailed fertilizer inputs for each treatment are summarized in Table 1. A complete randomized block and plot design with three replicate blocks was used. The area for each block was about 60 m$^2$ with 10 cm wide buffer zones on each side. Concrete frames (50 cm deep belowground and 15 cm aboveground) were installed to prevent water and nutrients exchange between the adjacent blocks. To meet the nutrient demands for rice growth, chemical fertilizers were added following expert recommendation for all treatments except for Control (Table 1). To reduce the risk of pest and disease, varieties of early and late rice were changed every 5 years.

**Table 1.** Details of fertilizer application in different treatments.

| Treatment | Early Rice (kg ha$^{-1}$ y$^{-1}$) | | | | Farmyard Manure | Late Rice (kg ha$^{-1}$ y$^{-1}$) | | | Milk Vetch Rice Straw |
|---|---|---|---|---|---|---|---|---|---|
| | Milk Vetch Manure | Chemical Fertilizers | | | | Chemical Fertilizers | | | |
| | | N | P$_2$O$_5$ | K$_2$O | | N | P$_2$O$_5$ | K$_2$O | |
| Control | - | - | - | - | - | - | - | - | - |
| M1 | 2250 | 45[a], 45[b], 69[c] | 30[a,b,c] | 0[a], 37.5[b], 75[c] | - | 45[a], 45[b], 69[c] | 30[a,b,c] | 0[a], 37.5[b], 75[c] | - |
| M2 | 2250 | 45[a], 45[b], 69[c] | 30[a,b,c] | 0[a], 37.5[b], 75[c] | 11,250 | 45[a], 45[b], 69[b] | 30[a,b,c] | 0[a], 37.5[b], 75[c] | 4050 |
| NPK | - | 90 | 45 | 75 | - | 90[a], 45[b], 69[c] | 60[a], 30[b], 30[c] | 0[a], 37.5[b], 75[c] | - |

[a, b, c] denoted chemical fertilizer amounts in 1981–1988, 1989–1995, and 1996–2017. The amount of organic amendments was based on their dried weights. The C content of milk vetch, farmyard manure, and rice straw were 402.5, 347.4, and 353.4 g kg$^{-1}$, respectively. The N content of milk vetch, farmyard manure, and rice straw were 20.51, 4.48, and 14.18 g kg$^{-1}$, respectively. The P content of milk vetch, farmyard manure, and rice straw were 22.87, 14.20, and 10.82 g kg$^{-1}$, respectively. The K content of milk vetch, farmyard manure, and rice straw were 10.80, 2.37, and 18.60 g kg$^{-1}$, respectively.

### 2.3. Crop Management

Rice seeds were sown in March for early rice and in June for late rice and transplanted into each plot at the densities of two seedling plant$^{-1}$ and 25 plants m$^{-2}$ for both early and late rice. Nitrogen, P, and K fertilizers were applied as urea (46.0% N), calcium magnesium phosphate (15.0% P$_2$O$_5$), and potassium chloride (60% K$_2$O), respectively. For both early and late rice, 30% of N, 30% of K, and total P were used as basal fertilizer, and the rest of the N and K fertilizers were applied as topdressing after 7 days of rice transplanting. The winter milk vetch of each plot was sown in the middle of October and directly rotary tilled into the soil in their own plots before early rice transplanting. Manure was incorporated into the soil during tillage before late rice planting. The late rice straw of each plot was crushed and then spread in their own plots on the surface. The crop residues of early rice and late rice were chopped and returned to the soil with rotary tillage after harvest. The irrigation regimes for both early and late rice followed local expert recommendation. Specifically, the water table level was maintained at about 5 cm aboveground from transplanting to tillering, then drained several times at booting and filling stage, and afterwards flooded with intermittent irrigation until about 10 days before harvest. Measures to control diseases, insect pests, and weeds followed the local agronomic practices. Grain yields were determined by harvesting the whole area of each plot.

### 2.4. Sample Collection

The soil samples were collected 7 days after late rice harvest in 2017. In each plot, soil samples of the plough layer (0–20 cm) were randomly collected from 5 points and mixed homogenously to produce a composite soil sample. Stones and residuals were then manually removed. Afterwards, the composite soil sample from each plot was divided into three parts. The first part was air-dried and then passed through a 2-mm sieve for soil chemical properties analysis. The second part was stored at 4 °C for microbial biomass C (MBC) and microbial biomass N (MBN). The last part was stored at −80 °C for the high-throughput sequencing of the bacterial community.

### 2.5. Measurement of Soil Properties

Soil pH was determined using a pH meter at a soil-to-water ratio of 1:2.5. The electrical conductivity (EC) was determined by a conductivity meter using a soil-to-water ratio of 1:5. The SOC, TN, available N, available P, and available K were determined by potassium dichromate external heating oxidation-volumetric, Kjeldahl digestion, alkali hydrolyzation-diusion, NaHCO$_3$ extraction–molybdenum antimony anti-colorimetric, CH$_3$COONH$_4$ extraction-flame photometry, respectively.

The NH$_4^+$ and NO$_3^-$ were extracted with 100 mL of 1 mol L$^{-1}$ KCl solution, shaken in a rotary shaker (140 r min$^{-1}$) for 1 h and filtered into sampling bottles. The solutions were analyzed by a colorimetric continuous flow analyzer (SANT++, Skalar Company, Breda,

The Netherlands). MBC and MBN were extracted by the chloroform fumigation method and determined with using a multi N/C 2100S CN analyser (Analytik Jena, Germany) [33].

*2.6. Measurement of Soil Bacterial Community*

Soil genomic DNA was extracted using the Power Soil DNA Isolation Kit (MO BIO Laboratories), following the manufacturer's protocol. Then, the DNA sample was stored at −80 °C until further processing. The V3-V4 region of the bacterial 16S rRNA gene was amplified with the primer 338F (5′-ACTCCTACGGGAGGCAGCA-3′) and 806R (5′-GGACTACHVGGGTWTCTAAT-3′) in two steps. Both the forward and reverse primers were tagged with adapter, pad, and link sequences.

Polymerase chain reaction (PCR) amplification was performed in a total volume of 50 μL reaction solution which contained 0.2 μL Q5 high-fidelity DNA polymerase, 10 μL high GC enhancer, 1 μL dNTP, 10 μM of each primer, and 60 ng genome DNA. PCR reactions were held at an initial denaturation at 95 °C for 5 min, followed by 15 cycles at 95 °C for 1 min, 50 °C for 1 min, and 72 °C for 1 min, with a final extension of 72 °C for 7 min. The PCR products from the first step were purified through VAHTSTM DNA clean beads. A second-round PCR was then performed in a 40 μL reaction solution, which contained 20 μL 2×Phμsion HF MM, 8 μL ddH2O, 10 μM of each primer, and 10 μL PCR products from the first step. Thermal cycling conditions were as follows: an initial denaturation at 98 °C for 30 s, followed by 10 cycles at 98 °C for 10 s, 65 °C for 30 s, and 72 °C for 30 s, with a final extension of 72 °C for 5 min. Finally, all PCR products were quantified by Quant-iT™ dsDNA HS Reagent and pooled together. The purified and pooled PCR samples were used for Illumina paired-end library construction, cluster generation, and high-throughput sequencing on the Illumina Hiseq 2500 platform (2 × 250 paired ends).

Raw tags were merged using the FLASH (V1.2.11) [34]. The merged sequences were then quality filtered by Trimmomatic (V 0.33), and chimera were removed by UCHIME (V 8.1) to achieve clean tags, which were clustered into OTU by USEARCH (V 10.0) at 97% similarity [35,36]. The OTU was filtered when relative abundance was less than 0.005%. A representative sequence was selected from each sample OTU, and the Ribosomal Database Project (RDP) classifier was used to assign taxonomic information. The alpha diversity was analyzed by Mothur (v.1.30) [37].

*2.7. Statistical Analysis*

All data analysis and plotting were performed in R 3.6.2 (https://www.r-project.org/ (accessed on 12 January 2023)). All original data used in this study are available on figshare (https://figshare.com/s/b9b1385825f9cc24196d (accessed on 12 January 2023)) and the Supplementary File. Yield stability was adopted to calculate the yield variations for each treatment [38,39], according to the following equation:

$$\text{Yield stability} = \frac{Mean_i}{SD_i} \tag{1}$$

where $SD_i$ and $Mean_i$ are the standard deviation and mean grain yield for each treatment across the period, respectively.

Before testing the treatment effects, all data were checked for a normal distribution based on the Shapiro–Wilk method and equality of variances using the Levene test at $p < 0.05$. Variables were log transferred when required. A linear mixed-effects model from the "*nlme*" package of R was used to investigate the effects of fertilization treatments on each observed variable. Regarding the long-term yield data, fertilization treatment, year, and their interactive effects are regarded as fixed effects, while block and plot are random factors. Differences between each treatment were evaluated using multiple comparison in the "*multcomp*" package of R. For yield stability across the 37 years and the other variables that were measured in 2017, fertilization treatment is considered the fixed effect, while the plot and block are the random factors. Mixed regression analysis was adopted to study the links between grain yield, yield stability, and all other biotic and abiotic factors,

considering the plot and block as random factors. The R-squared is estimated based on the "*r.squaredGLMM*" function from the R package "*MuMIn*". All residuals were checked for normality. In addition, a Mantel test was performed to examine the associations between the environmental variables and bacterial communities.

## 3. Results

### *3.1. Rice Grain Yield and Yield Stability*

Averaged across the 37-year period, NPK, M1, and M2 significantly increased rice grain yield by 54, 46, and 73% compared to Control, respectively (Figure 1A). Apart from Control, differences in grain yield were also observed between fertilization treatments. Specifically, M2 significantly increased grain yield by 18% when compared to M1 and by 12% when compared to NPK. Regarding yield stability, it was increased significantly by 22, 17, and 9% in NPK, M1, and M2, respectively (Figure 1B). There was no significant difference for yield stability between M1 and NPK, while they had larger yield stability than M2 by 7 and 12%, respectively. There were also significant effects of fertilization treatment, year, and their interactive effects on annual rice grain yield (Table S1). The positive effects of fertilization treatments on rice grain yield were true even when evaluated separately in each year (Figure 1C and Table S1).

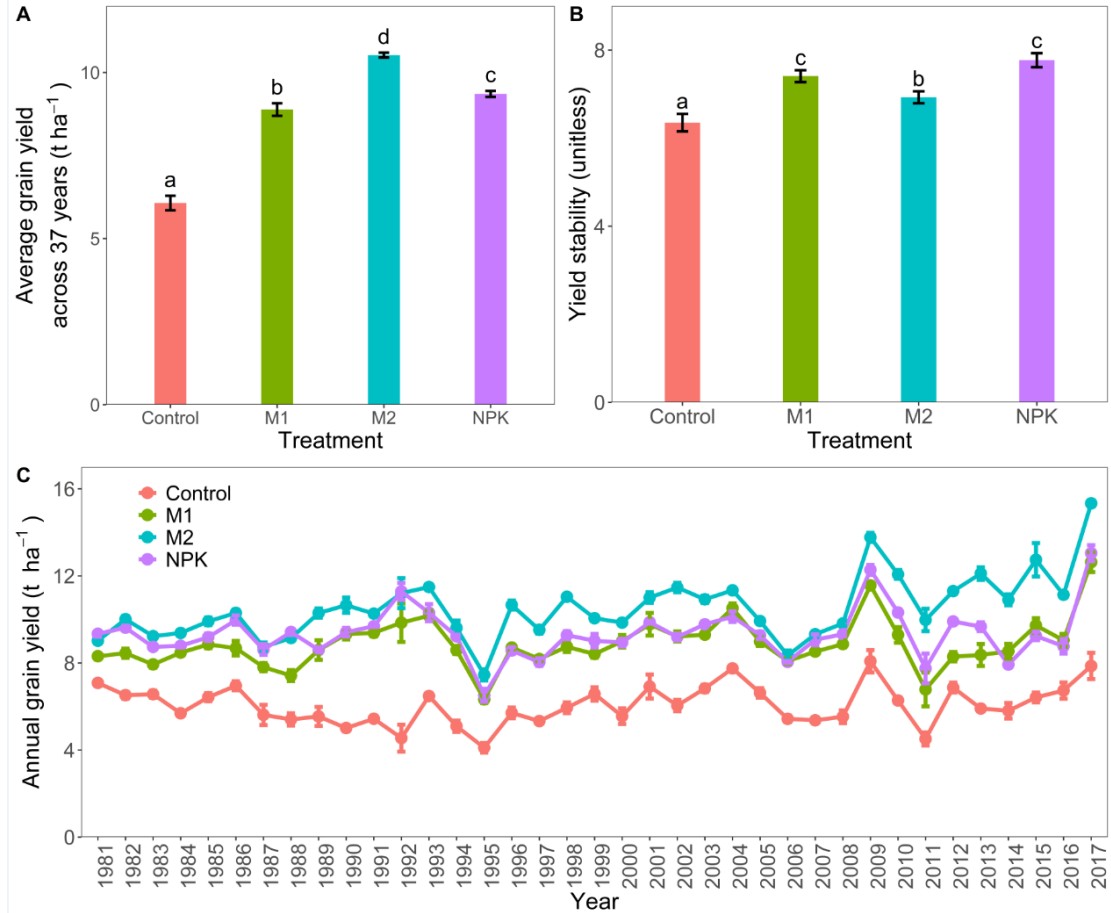

**Figure 1.** The effects of long-term mineral and manure fertilization on (**A**) average rice grain yield and (**B**) yield stability across the 37 years. The effects of long-term mineral and manure fertilization on (**C**) annual rice grain yield. There were four treatments: no application of fertilizer (Control); application of nitrogen–phosphorus–potassium fertilizer in early rice (NPK); NPK plus green manure in early rice (M1); and NPK plus green manure in early rice and farmyard manure in late rice and rice straw return in winter (M2). Values are mean ± standard errors of four replicates. Values without shared letters indicate significant difference at $p < 0.05$.

### 3.2. Soil pH, EC and Nutrient Availability

Soil pH varied from 4.98 to 5.34. M1 and M2 significantly increased soil pH by a unit of 0.28 and 0.30 compared to Control, respectively (Figure 2A). There was no significant difference on soil pH between Control and NPK. Regarding soil EC, it was significantly increased in M2 by 84% compared to Control, whereas there were no significant differences between Control, M1, and NPK (Figure 2B).

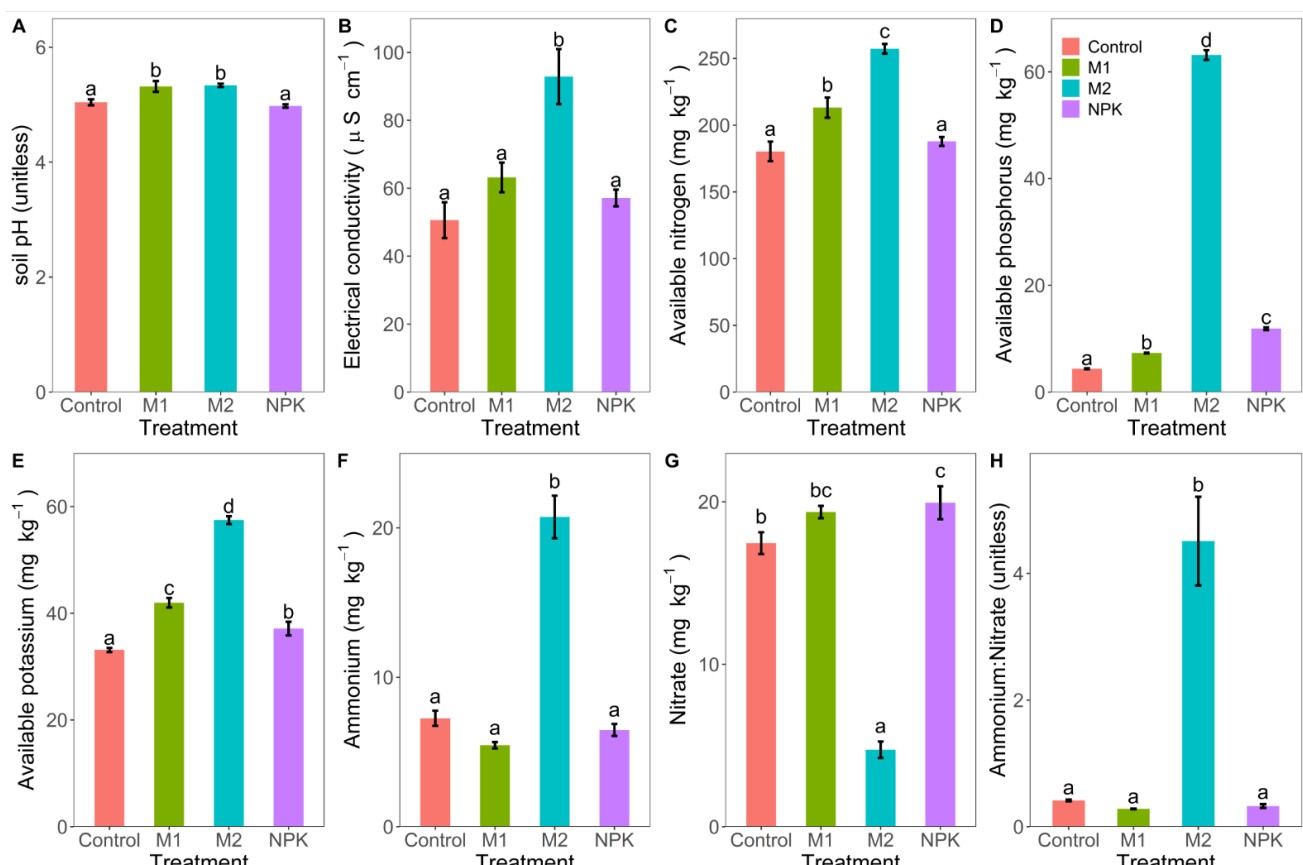

**Figure 2.** The effects of long-term mineral and manure fertilization on (**A**) soil pH, (**B**) soil EC, (**C**) soil available nitrogen, (**D**) soil available phosphorus, (**E**) soil available potassium, (**F**) soil ammonium, (**G**) soil nitrate, and (**H**) the ratio of ammonium:nitrate. There were four treatments: no application of fertilizer (Control); application of nitrogen–phosphorus–potassium fertilizer in early rice (NPK); NPK plus green manure in early rice (M1); and NPK plus green manure in early rice and farmyard manure in late rice and rice straw return in winter (M2). Values are mean $\pm$ standard errors of four replicates. Values without shared letters indicate significant difference at $p < 0.05$.

Compared to control, M1 and M2 significantly increased soil available N content by 18 and 43%, respectively, but there was no difference for NPK (Figure 2C). M2 also significantly increased soil available N content by 21% compared to M1. Regarding soil available P content, it significantly increased in NPK, M1, and M2 by 172, 68, and 1347% compared to Control, respectively (Figure 2D). For soil available K content, it significantly increased in NPK, M1, and M2 by 12, 27, and 73%, respectively (Figure 2E). A significantly higher soil available K content was also observed for M2 compared to M1 and NPK.

M2 significantly increased soil $NH_4^+$ content by 186% compared to Control, whereas there was no significant difference between Control, M1 and NPK (Figure 2F). On the contrary, M2 significantly decreased soil $NO_3^-$ content by 73% compared to Control, while it significantly increased by 14% in NPK compared to Control (Figure 2G). The contrasting effects of M2 on soil $NH_4^+$ and $NO_3^-$ content led to a substantially increased $NH_4^+$:$NO_3^-$

ratio compared to Control, whereas there was no difference for the $NH_4^+:NO_3^-$ ratio between Control, M1, and NPK (Figure 2H).

### 3.3. Microbial Biomass Carbon and Nitrogen, Soil Organic Carbon and Total Nitrogen

M1 and M2 significantly increased MBC by 104 and 52% compared to Control, respectively, whereas NPK had no effect (Figure 3A). Regarding MBN, it significantly increased by 36% in M2 relative to Control, whereas there were no differences between Control, M1 and NPK (Figure 3B). For MBC:MBN, it significantly increased by 52% in M2 relative to Control, whereas there were no differences between Control, M1, and NPK (Figure 3C).

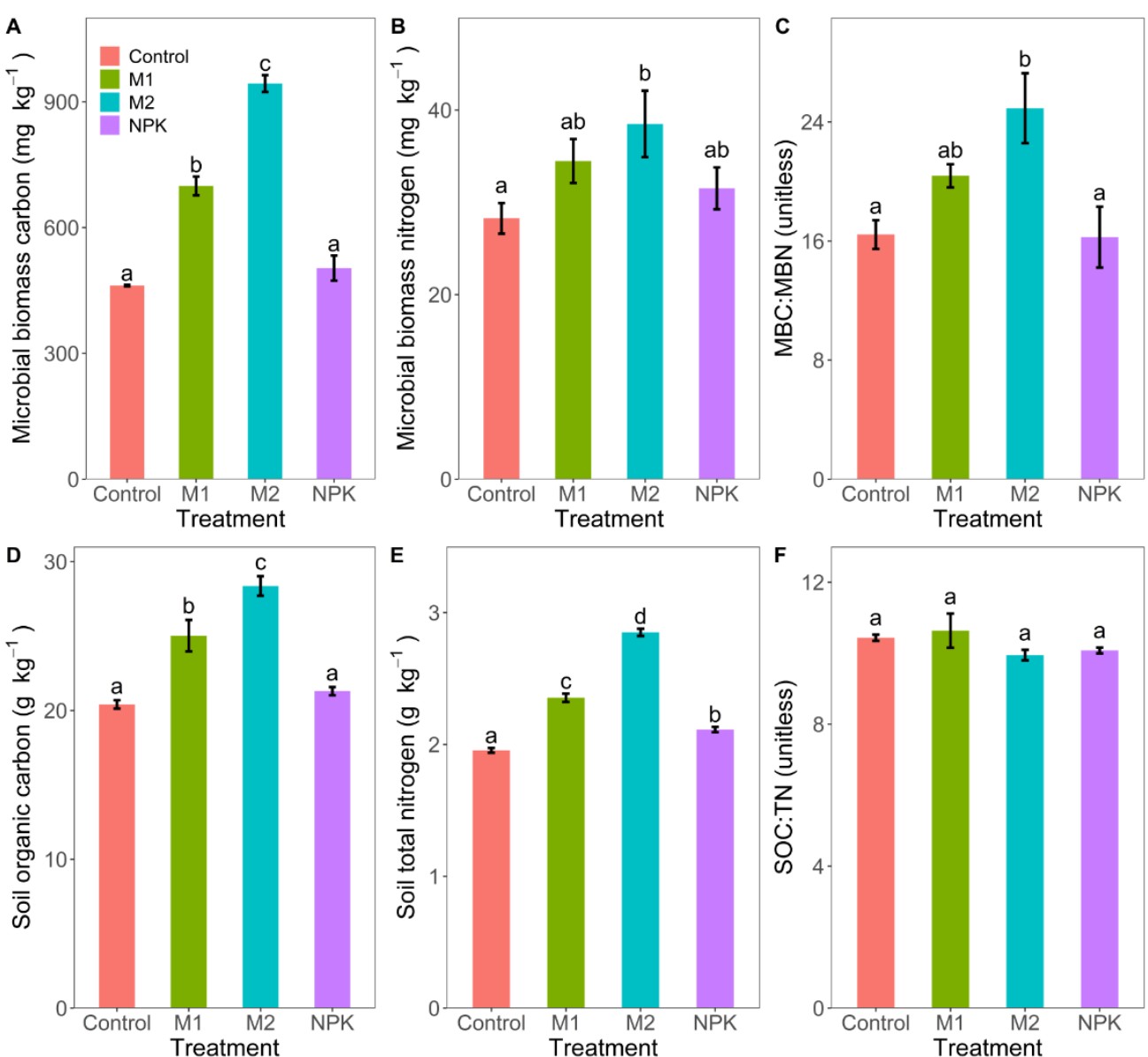

**Figure 3.** The effects of long-term mineral and manure fertilization on (**A**) microbial biomass carbon (MBC), (**B**) microbial biomass nitrogen (MBN), (**C**) MBC:MBN, (**D**) soil organic carbon (SOC), (**E**) soil total nitrogen (TN), and (**F**) SOC:TN. There were four treatments: no application of fertilizer (Control); application of nitrogen–phosphorus–potassium fertilizer in early rice (NPK); NPK plus green manure in early rice (M1); and NPK plus green manure in early rice and farmyard manure in late rice and rice straw return in winter (M2). Values are mean ± standard errors of four replicates. Values without shared letters indicate significant difference at *p* < 0.05.

M1 and M2 significantly increased SOC content by 39 and 23% compared to Control, respectively, whereas there was no difference between Control and NPK (Figure 3D). Regarding soil TN content, it significantly increased by 8, 46 and 20% in NPK, M1 and M2 compared to Control, respectively (Figure 3E). In addition, NPK, M1 and M2 had no effect on soil C:N (Figure 3F).

### 3.4. Bacterial Community Diversity and Composition

M2 significantly increased the number of OTU by 68% compared to Control, whereas there were no significant differences between Control, M1, and NPK (Figure 4A). Similarly, M2 significantly increased the Chao1 by 79% compared to Control, whereas there were no significant differences between Control, M1, and NPK (Figure 4B). There was no difference for Shannon index between different fertilization treatments (Figure 4C). M1 significantly decreased Simpson index by 10%, whereas there were no significant differences between Control, M2, and NPK (Figure 4D).

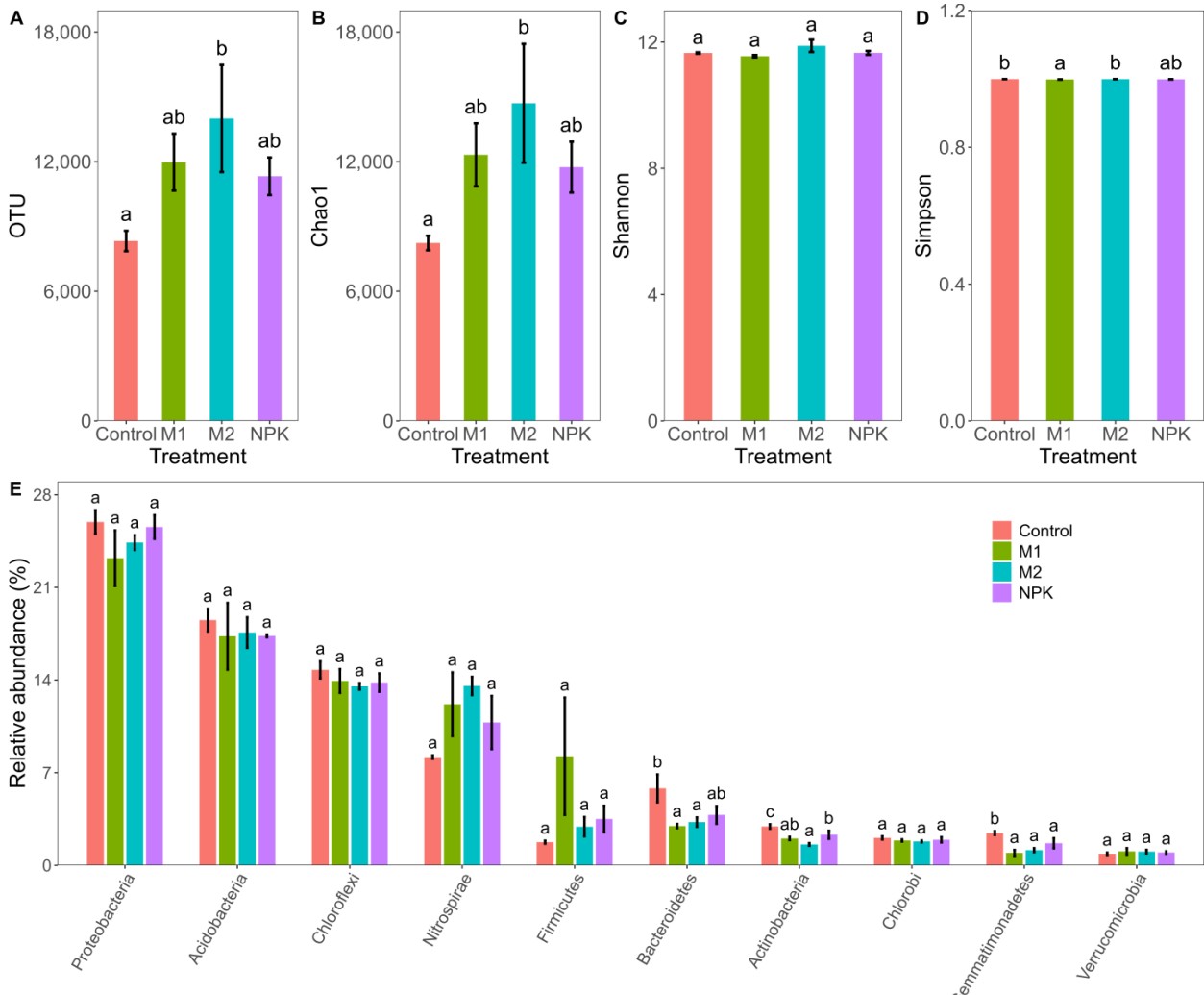

**Figure 4.** The effects of long-term mineral and manure fertilization on (**A**) the number of bacterial OTU, (**B**) Chao1, (**C**) Shannon index, (**D**) Simpson index, and (**E**) bacterial community composition. There were four treatments: no application of fertilizer (Control); application of nitrogen–phosphorus–potassium fertilizer in early rice (NPK); NPK plus green manure in early rice (M1); and NPK plus green manure in early rice and farmyard manure in late rice and rice straw return in winter (M2). Values are mean ± standard errors of four replicates. Values without shared letters indicate significant difference at *p* < 0.05.

The dominant bacterial phyla (average relative abundance more than 1%) accounted for 81–84% of the total OTU. They were Proteobacteria, Acidobacteria, Chloroflexi, Nitrospirae, Firmicutes, Bacteroidetes, Actinobacteria, Chlorobi, Gemmatimonadetes, Verrucomicrobia (Figure 4E). M1 and M2 significantly decreased Bacteroidetes abundance by 44 and 49% compared to Control, respectively. For Actinobacteria abundance, it significantly decreased by 6, 7, and 5% in NPK, M1, and M2 compared to Control, respectively. Regarding Gemmatimonadetes abundance, it significantly decreased by 32, 61, and 53% in NPK, M1, and M2 compared to Control, respectively. However, there was no significant fertilization treatment effects on other phyla.

### 3.5. Factors Affecting Grain Yield, Yield Stability, and Bacterial Community

The correlation analysis showed that mineral- and manure-induced changes in rice grain yield were positively correlated with soil available nutrient content, soil TN, soil EC, $NH_4^+$, MBC, and bacterial Simpson index but negatively correlated with $NO_3^-$ and Actinobacteria abundance (Figures 5 and S5). Meanwhile, mineral- and manure-induced increases in MBC were positively correlated with the corresponding changes in SOC and TN content (Figures 6 and S5). In addition, mineral- and manure-induced changes in SOC and TN content were negatively correlated with the changes in yield stability (Figures 7 and S5).

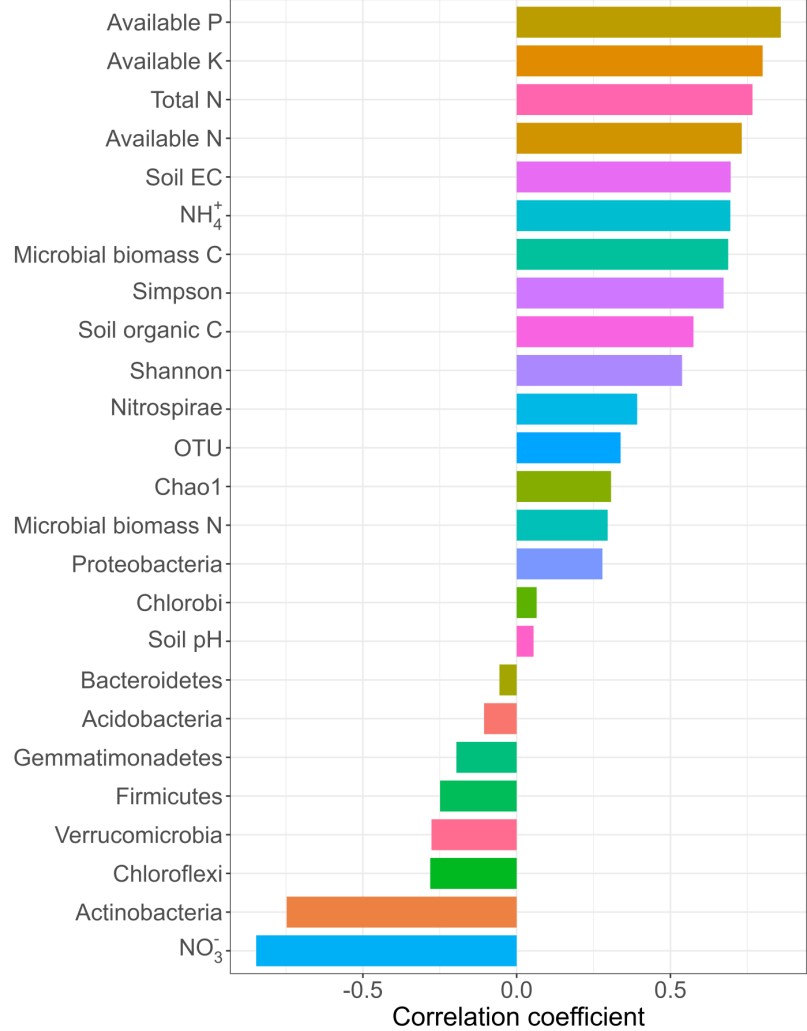

**Figure 5.** Correlation analysis between mineral- and manure-induced changes in rice grain yield and the corresponding changes in other studied variables.

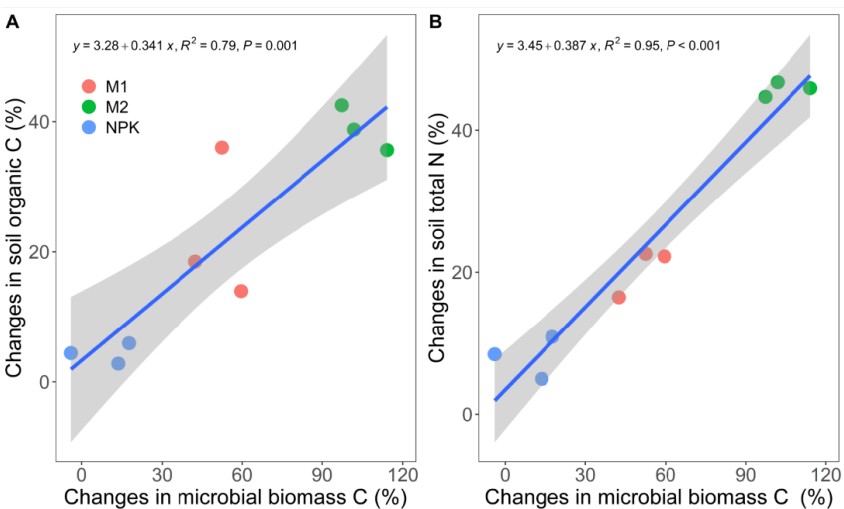

**Figure 6.** Relationships between mineral- and manure-induced changes in (**A**) soil organic carbon, (**B**) soil total nitrogen and the corresponding changes in microbial biomass carbon. Application of nitrogen–phosphorus–potassium fertilizer in early rice (NPK), NPK plus green manure in early rice (M1), NPK plus green manure in early rice and farmyard manure in late rice and rice straw return in winter (M2).

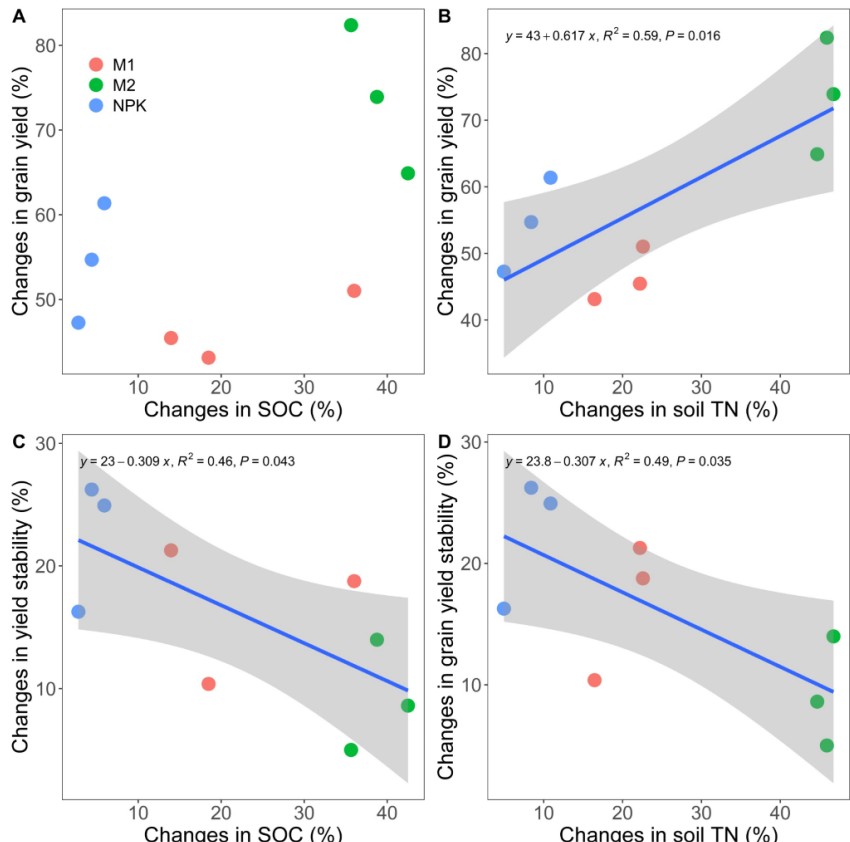

**Figure 7.** Relationships between mineral- and manure-induced changes in rice grain yield and the corresponding changes in (**A**) soil organic carbon (SOC) and (**B**) soil total content (TN). Relationships between mineral- and manure-induced changes in rice yield stability and changes in (**C**) SOC and (**D**) TN. Application of nitrogen–phosphorus–potassium fertilizer in early rice (NPK), NPK plus green manure in early rice (M1), NPK plus green manure in early rice and farmyard manure in late rice and rice straw return in winter (M2).

## 4. Discussion

### 4.1. Increased Rice Grain Yield and Yield Stability with Long-Term Fertilization

Long-term mineral (NPK) and manure (M1 and M2) fertilization have significantly increased rice grain yield (Figure 1). Our results are in line with other studies that showed positive effects of fertilization on rice grain yield [40,41]. Intensive harvesting of the double rice-cropping system likely results in the depletion of soil nutrients [42], and it is not surprising that both the mineral and manure fertilization treatments had pronounced positive effects on rice grain yield in the experimental double rice-cropping system. This explanation is confirmed by our correlation analysis that grain yield was positively correlated with soil nutrient availability (Figure 5). Some studies have reported that long-term fertilization reduced grain yield due to soil acidification [6,7]. In this study, mineral addition had no effect on soil pH, and we even observed increases in soil pH after manure fertilization. Thus, changes in soil pH may not necessarily limit the effects of mineral and manure fertilization on rice grain yield, at least not in this study site. In addition, fertilization-induced increases in bacterial Simpson index were positively correlated with changes in rice grain yield. These results highlight the potential linking the shifts in bacterial community with grain yield, despite the challenges in identifying the specific soil microorganisms. Regarding manure fertilization, M2 had significantly larger positive effects on rice grain yield than M1. One explanation might be associated with the higher soil pH and soil EC under M2, which may help stimulating soil microbial activity and maintaining high soil nutrient availability [8,43,44]. Another explanation might be related to the increased soil $NH_4^+$ and decreased soil $NO_3^-$ content under M2 since rice prefer $NH_4^+$ more than $NO_3^-$ in wetter conditions [45]. In addition, the farmyard manure and the associated macro- and micro-nutrients with M2 might also contribute to the increased grain yield [46].

Mineral and manure fertilization have profoundly increased yield stability across the 37-year period (Figure 1). Although yield stability from the double rice-cropping system was not sufficiently reported, our results agree with studies from other rice cropping systems that showed that fertilization significantly increased rice yield stability [9,10]. Additional nutrient inputs may reduce plant resource investment for nutrient acquisition, which may increase rice adaptation to unfavorable conditions [47–49], such as drought or temperature stress [50]. This explanation is corroborated by the absence of relationships between fertilization-induced changes in rice grain yield and annual temperature and precipitation over the 37 years (Figures S3 and S4). Another explanation might be related to the shifts in soil microbial community composition since we observed clear positive correlations between fertilization-induced changes in Actinobacteria and Chlorobi abundances and the corresponding changes in yield stability. Despite the fact that the direct evidence linking Actinobacteria and Chlorobi abundances with yield stability is lacking, the emerging recent studies have demonstrated that soil microorganisms played pivotal roles in ecosystem adaptation to environmental stresses and nutrient deficits [26,27,51]. The divergent responses of grain yield and yield stability to mineral and manure fertilization suggest that it may be necessary to consider them separately when designing sustainable double rice-cropping systems in the future. Nevertheless, due to the limited number of long-term studies on the effects of mineral and manure fertilization on yield stability, the driving mechanisms are still under discussion.

### 4.2. Enhanced Soil Organic Carbon and Total Nitrogen Content with Long-Term Fertilization

Long-term manure fertilization (M1 and M2) significantly increased SOC and TN content more than mineral fertilization (Figure 5). Our results indicate the importance of organic matter inputs in increasing SOC and TN content, which are in line with several other studies [52,53]. First, manure application and the associated additional C and N inputs, particularly for the straw incorporation in M2, will directly contribute to SOC and TN accumulation [40]. This explanation was supported by the non-significant effect of mineral addition on SOC, which was likely due to the absence of organic matter inputs. Second, increases in MBC and MBN may contribute to SOC and TN accumulation since C

and N secured in microbial biomass are reported to be more stable [54]. This contention was corroborated by (1) the more pronounced positive effects of M1 and M2 on MBC and MBN and (2) fertilization-induced changes in SOC and TN were positively correlated with the changes in MBC and MBN. Third, manure fertilization will improve a range of soil physical variables, such as soil bulk density, soil aggregates, soil porosity, and soil water holding capacity, which may directly and indirectly contribute to SOC and TN accumulation [49,55–57]. In addition, mineral and manure inputs may either stimulate or suppress soil organic matter decomposition and therefore affect SOC and TN dynamics [52], which should be further investigated in future research.

There was no clear relationship between fertilization-induced changes in SOC and rice grain yield, suggesting the possibility of increasing rice grain yield without decreasing SOC content [40,54]. For example, mineral addition significantly increased rice grain yield but had no effect on SOC content. It has been hypothesized that increases in grain yield and associated C allocation belowground would directly increase SOC content. However, the lack of a relationship between changes in grain yield and SOC in this study suggests that the indirect effects of increased grain yield on SOC are likely to be larger than the direct effects of plant residual inputs, for example, by changing microbial community composition and metabolic activity [58,59]. This explanation has been confirmed by the increased OTU and Chao1 with M2, which were accompanied with increased SOC content. On the contrary, negative relationships were observed between fertilization-induced changes in SOC, TN, and yield stability, which could possibly be due to the unaccompanied large increases in yield stability after M2, despite the underlying mechanisms still being unclear. These contrasting relationships between SOC, TN, yield, and yield stability highlight the necessity to fully integrate rice grain yield, yield stability, and SOC and TN content into new conceptual frameworks supporting the design of sustainable double rice-cropping systems.

*4.3. Shifts in Bacterial Community*

Our results on bacterial community were consistent with other studies that showed that organic manure inputs profoundly increased the microbial richness [60,61]. However, this conclusion only held true for M2, which had the most complex mix of organic manure inputs in this study. Firstly, M2 had more pronounced positive effects on soil pH and soil EC, which may help creating favorable conditions for microbial growth and proliferation [62]. This contention is in line with other studies that showed that improvements of soil biotic and abiotic factors can have cascading effects on microbial growth and proliferation [8,44]. Secondly, the return of milk vetch and rice straw with M2 provides readily accessible resources to support microbial growth and proliferation, as straw incorporation has been shown to significantly increase microbial richness and abundance [60,61]. Thirdly, M2 significantly increased MBN, which were positively correlated with OTU richness as revealed by both the Mantel test and correlation analysis. Thus, the microbial growth and proliferation are likely primarily limited by N availability, leading to the observed substantial microbial responses with the external N inputs.

Both mineral and manure fertilization significantly decreased the relative abundance of Actinobacteria and Gemmatimonadetes, and M1 and M2 also decreased the relative abundance of Bacteroidetes. Our results supported previous observations of shifts in microbial community composition from external fertilizer inputs [58,60,61]. Indeed, Actinobacteria, Gemmatimonadetes, and Bacteroidetes have been identified as oligotrophic bacteria, and their abundance declines with increasing soil nutrient availability [63,64]. The underlying mechanisms associated with shifts in bacterial community composition are largely unclear and differ widely across various study sites [60,61]. For example, changes in Bacteroidetes abundance were negatively correlated with MBN, while Actinobacteria abundance decreased with the increasing soil nutrient availabilities. Some studies have shown that changes in microbial community composition might also be driven by nutrient acquisition or stress tolerance [65,66], which might be fundamentally associated with

changes in microbial physiology. However, changes in microbial physiology after mineral and manure fertilization are beyond the scope of this study.

### 4.4. Changes in Soil Physicochemical Variables

Manure fertilization (M1 and M2) significantly increased soil pH and M2 also significantly increased soil EC, suggesting the advantages of manure inputs in improving soil pH and EC. The alkalinity and cations of plant materials incorporated into soil can directly increase soil pH and EC [67]. Manure fertilization had more pronounced positive effects than mineral fertilization on soil available N, P, and K content. Plant residue inputs with manure fertilization will secure soil available nutrients by forming and stabilizing macroaggregates as well as by reducing soil nutrients leaching [68–70]. In addition, milk vetch, manure, and rice straw can also serve as C and N resources to support microbial growth, proliferation, and metabolic activity, which will primarily contribute to the increased MBC and MBN [71,72].

### 4.5. Uncertainties and Implications

There are several limitations and uncertainties that should be noted when interpretating our results. First, the soil microbial community and a range of other reported soil variables were only measured once in 2017, 37 years after the initiation of the fertilization experiments. Caution is required when comparing these results with results from short-term experiments since some variables likely respond differently to short- and long-term experimental durations [73–75]. Second, it is challenging to clarify the causal relationships between rice grain yield, yield stability, soil physical and chemical variables, and changes in soil microbial community composition due to the strong mutual and interactive effects among these processes. However, our results significantly advance the understanding of the potential links between them and provide new perspectives for future research priorities. Third, we primarily focused on soil bacterial communities, while soil fungi or other functional microorganisms were not included in this study, resulting in an incomplete picture of shifts in soil microbial community composition and their relationships with other investigated variables [76]. Further integration of state-of-the-art microbial functional gene abundance methods and advanced statistical analyses is therefore recommended in future studies.

## 5. Conclusions

Our results demonstrate the advantages of organic manure fertilization on grain yield, yield stability, SOC, and TN content over decades in double rice-cropping systems. Increases in soil pH, soil EC, soil available nutrient content, and MBC and MBN are positive correlated with the changes in grain yield, SOC, and TN content, although more research is required to clarify the causal relationships between them. Increases in SOC and TN content are positively correlated with the long-term averaged rice grain yield, whereas negative relationships are observed for yield stability. These contrasting relationships highlight the importance of simultaneously considering grain yield, yield stability, SOC, and TN content when developing sustainable double rice-cropping systems. In addition, there are significant correlations between shifts in bacterial community and the changes in grain yield and yield stability, underscoring the necessity for more in-depth research of soil microbiological parameters under these management scenarios. Overall, this systematic investigation of grain yield, yield stability, SOC, and TN content and their underlying mechanisms over long-term (37 years) double rice-cropping practices provides new perspectives for developing sustainable and intensive rice cropping systems to meet the growing global demand for rice production.

**Supplementary Materials:** The following supporting information can be downloaded at: https://www.mdpi.com/article/10.3390/agronomy13010261/s1.

**Author Contributions:** J.L. (Jin Li), K.-L.L., J.X., G.-Q.D., D.-M.L., X.-J.G., X.-H.L., X.-M.C., C.-F.Q., Y.-F.Q., W.-J.X. and J.C. (Jin Chen): Data curation, Formal analysis, Validation; J.L. (Jin Li), K.-L.L., J.C. (Jin Chen) and Y.J.: Conceptualization, Investigation; J.C. (Jin Chen) and J.C. (Ji Chen): Funding acquisition; J.L. (Jin Li), J.L. (Jia Liu), J.C. (Ji Chen) and J.C. (Jin Chen): Software, Visualization, Writing—original draft; J.L. (Jin Li), J.C. (Ji Chen), J.C. (Jin Chen), Y.J., C.-R.P. and S.M.B.: Writing—Reviewing and Editing. All authors have read and agreed to the published version of the manuscript.

**Funding:** This research was funded by National Natural Science Foundation of China (32060431), National Key Research and Development Program of China (2016YFD0300901), Aarhus Universitets Forskningsfond (AUFF-E-2019-7-1), EU H2020 Marie Skłodowska-Curie Actions (839806), Danish Independent Research Foundation (1127-00015B), Nordic Committee of Agriculture and Food Re-search (https://nordicagriresearch.org/2020-5/ (accessed on 12 January 2023)), Science and Technology Project of Jiangxi Provincial Education Department (GJJ191138), and Doctoral Research Fund of Nanchang Normal University (090170003312). The APC was funded by Aarhus University.

**Data Availability Statement:** The data associated with this paper is available from the online supplementary file or from the figshare (https://figshare.com/s/b9b1385825f9cc24196d (accessed on 12 January 2023)).

**Conflicts of Interest:** The authors declare that they have no known competing financial interest or personal relationship that could have appeared to influence the work reported in this paper.

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
