# Peer review of "New Insights from Soil Microorganisms for Sustainable Double Rice-Cropping System with 37-Year Manure Fertilization"

_agronomy, doi:10.3390/agronomy13010261_

Round 1

Reviewer 1 Report

The study compared the effects of NPK, M1 and M2 treatments on rice yield, soil carbon and nitrogen chemical indicators, and microbial carbon and nitrogen biological indicators, etc. I think this article is seriously lacking in innovation (just simple correlation and linear fitting), and the only advantage is that the field test period is as long as 37 years. The conclusions of the studies are known and widely reported, and the complex relationships between the indicators are not revealed. I don't even know which measure (M1 or M2?) is the most promising for rice production in Jiangxi province. I do not recommend acceptance of this manuscript.

Some other small suggestions

1. Plain and long title

2. Irregular keywords, especially the last two.

3. Line 47. First?? I'm pretty sure you're exaggerating.

4. What does hill mean? In Line 172.

5. Line 250. What is the difference between ‘yield stability’ and the ‘coefficient of variation’ of yield? I think there is no need for this definition. In addition, does this indicator need a unit?

Author Response

The study compared the effects of NPK, M1 and M2 treatments on rice yield, soil carbon and nitrogen chemical indicators, and microbial carbon and nitrogen biological indicators, etc. I think this article is seriously lacking in innovation (just simple correlation and linear fitting), and the only advantage is that the field test period is as long as 37 years. The conclusions of the studies are known and widely reported, and the complex relationships between the indicators are not revealed. I don't even know which measure (M1 or M2?) is the most promising for rice production in Jiangxi province. I do not recommend acceptance of this manuscript.

[Response] Thank you very much for your instructive comments and suggestions. Please see below our point-by-point responses. 

Some other small suggestions

  1. Plain and long title

[Response] Modified (Line 2-3). The new title reads, “New insights from soil microorganisms for sustainable double rice-cropping system with 37-year manure fertilization”.

  1. Irregular keywords, especially the last two.

[Response] They are replaced by “soil nutrient” and “sustainable agriculture”, respectively (Line 52).

  1. Line 47. First?? I'm pretty sure you're exaggerating.

[Response] Modified (Line 45).

  1. What does hill mean? In Line 172.

[Response] It means 25 plants per square meter, which is a common word used in rice cropping system.

  1. Line 250. What is the difference between ‘yield stability’ and the ‘coefficient of variation’ of yield? I think there is no need for this definition. In addition, does this indicator need a unit?

[Response] We would like politely argue that they are different. Yield stability means the variation over time (across the 37 years). Coefficient of variation indicates the variations for each treatment.

Reviewer 2 Report

The topic of this work is interesting, and the experimental design is well structured, the choice of rice is interesting for its importance as food crop and for the particular kind of cultivation (double rice-cropping system) that would allow to improve grain yield. The choice to study the structural bacterial biodiversity by the massive sequencing (NGS) is appropriate and provide a good description of the soil bacterial community. The work, in general, is not so innovative both for the kind of analyses performed and for the alternative soil treatments proposed, moreover, as reported by the authors in the 4.5 section, there are several limitations such as the analyses performed only once in 2017 or the lack of a complete soil microbial community analysis which also include fungi. But this work has its strength in the duration of the treatments, it’s not so common to have an experimental field with 37 years of certified treatments and this is really important especially to evaluate the effects of different treatments on soil properties. For this reason, I think that this manuscript is interesting and I hope that the authors will perform more in-depth analyses in the future.

In general, the manuscript is well written and there aren’t significant modifications to suggest.

Below I reported some minor suggestions about specific parts of the paper:

Results

Line 289: the authors can explain better with respect to which samples the increases in values were recorded

Line 316-317: looking at the graph, it does not seem that there is actually an increase of 68% of OTU in M2 sample compared to Control, the authors should check the data

Discussion

Line 358-360: I don’t understand what the authors mean with this sentence, they never consider the effects of plant-growth-promoting isolation or use, and they don’t report any reference, so they have to explain better what they want to say or delete the sentence.

Line 378-384: the authors suppose that fertilization-induced changes in Actinobacteria and Chlorobi abundances could be related to changes to yield stability and support this hypothesis citing recent studies that have demonstrated that “soil microorganisms play pivotal roles on ecosystem adaptation to environmental stresses and nutrient deficits” but it’s not so clear if these studies report the role of Actinobacteria and Chlorobi. Please explain better.

Author Response

The topic of this work is interesting, and the experimental design is well structured, the choice of rice is interesting for its importance as food crop and for the particular kind of cultivation (double rice-cropping system) that would allow to improve grain yield. The choice to study the structural bacterial biodiversity by the massive sequencing (NGS) is appropriate and provide a good description of the soil bacterial community. The work, in general, is not so innovative both for the kind of analyses performed and for the alternative soil treatments proposed, moreover, as reported by the authors in the 4.5 section, there are several limitations such as the analyses performed only once in 2017 or the lack of a complete soil microbial community analysis which also include fungi. But this work has its strength in the duration of the treatments, it’s not so common to have an experimental field with 37 years of certified treatments and this is really important especially to evaluate the effects of different treatments on soil properties. For this reason, I think that this manuscript is interesting and I hope that the authors will perform more in-depth analyses in the future. In general, the manuscript is well written and there aren’t significant modifications to suggest. Below I reported some minor suggestions about specific parts of the paper:

[Response] Thank you very much for your positive assessments. Please see below our point-by-point responses. 

Results

Line 289: the authors can explain better with respect to which samples the increases in values were recorded

[Response] Updated (Line 287-289).

Line 316-317: looking at the graph, it does not seem that there is actually an increase of 68% of OTU in M2 sample compared to Control, the authors should check the data

[Response] We have double-checked the data. It is correct. Please note it is the proportional change.

Discussion

Line 358-360: I don’t understand what the authors mean with this sentence, they never consider the effects of plant-growth-promoting isolation or use, and they don’t report any reference, so they have to explain better what they want to say or delete the sentence.

[Response] Thanks for this great comment. This sentence has been modified (Line 357-358). The new sentence reads “These results highlight the potential linking the shifts in bacterial community with grain yield, despite the challenges in identifying the specific soil microorganisms”.

Line 378-384: the authors suppose that fertilization-induced changes in Actinobacteria and Chlorobi abundances could be related to changes to yield stability and support this hypothesis citing recent studies that have demonstrated that “soil microorganisms play pivotal roles on ecosystem adaptation to environmental stresses and nutrient deficits” but it’s not so clear if these studies report the role of Actinobacteria and Chlorobi. Please explain better.

[Response] A great point. We are just starting to do more research in this area, whereas the direct evidence linking Actinobacteria and Chlorobi abundances with yield stability is lacking. We have modified the writing (Line 380-383). The new sentence reads “Despite the direct evidence linking Actinobacteria and Chlorobi abundances with yield stability is lacking, the emerging recent studies have demonstrated that soil microorganisms played pivotal roles on ecosystem adaptation to environmental stresses and nutrient deficits”.

Round 2

Reviewer 1 Report

The response avoid the important and dwell on the trivial. However, the response to small questions are also not professional. 

Science requires both intradisciplinary and interdisciplinary communication. Your paper is not just for people who study rice. It is best to make a note to let readers know what the density (plant ha-1) is. Add '(?? plant ha-1, approximately)' after hill.

Does the coefficient of variation only refer to the degree of dispersion between treatments? Your so-called 'stability' is nothing more than the reciprocal of the coefficient of variation of annual yield. It's actually the same concept. The former is bigger and more stable, and the latter is smaller and more stable. I think this indicator has no value at all, it's just creating data in different ways. Although some people do this, it does not mean that it is an indicator worth calculating.

What's more, the most critical question of innovation has not been answered and highlighted. I know the long-term positioning experiment is valuable, but that doesn't make this draft worthwhile. That said, I don't think this draft offers a new perspective on rice cultivation or new insights in the future. Because the results all you reported are generally known. Please don't avoid my core question unless proven to me. It doesn't matter, welcome rebuttal.

Author Response

Reviewer #1

The response avoid the important and dwell on the trivial. However, the response to small questions are also not professional.

[Response] Thanks for your useful comments. We are sorry for some unexpected misunderstanding. Please see our point-by-point responses below.

Science requires both intradisciplinary and interdisciplinary communication. Your paper is not just for people who study rice. It is best to make a note to let readers know what the density (plant ha-1) is. Add '(?? plant ha-1, approximately)' after hill.

[Response] We agree with you. It was replaced by plant ha-1.

Does the coefficient of variation only refer to the degree of dispersion between treatments? Your so-called 'stability' is nothing more than the reciprocal of the coefficient of variation of annual yield. It's actually the same concept. The former is bigger and more stable, and the latter is smaller and more stable. I think this indicator has no value at all, it's just creating data in different ways. Although some people do this, it does not mean that it is an indicator worth calculating.

[Response] Please let me politely explain the difference again.  Yield stability means the variation over time (across the 37 years), whereas coefficient of variation indicates the variations for each treatment (variations relative to mean values). Both variables are very important to understand the yield variations at different scales. Please see a very much relevant reference here (Knapp et al., 2018). You are welcomed to contact me if you still have some difficulties to understand the differences between them.

Knapp S., Van Der Heijden M. G. A. (2018) A global meta-analysis of yield stability in organic and conservation agriculture. Nature communications, 9 (1), 3632. https://doi.org/10.1038/s41467-018-05956-1

What's more, the most critical question of innovation has not been answered and highlighted. I know the long-term positioning experiment is valuable, but that doesn't make this draft worthwhile. That said, I don't think this draft offers a new perspective on rice cultivation or new insights in the future. Because the results all you reported are generally known. Please don't avoid my core question unless proven to me. It doesn't matter, welcome rebuttal.

[Response] Thanks for your useful comments. We at least have the following novelty, (1) the simultaneously documentation of yield, yield stability, and soil C and N content with long-term organic and inorganic fertilization, (2) the novel perspective from soil microbiology, (3) the new double rice-cropping system, and (4) the long-term experimental platform. This study provides pioneer comprehensive assessments of the simultaneous responses of grain yield, yield stability, SOC and TN content, nutrient availability and bacterial community com-position to long-term mineral and manure fertilization in a double rice-cropping system. Altogether, this study spanning nearly four decades provides new perspectives for developing sustainable yet intensive rice cultivation to meet growing global demands.
